# Optimal Techniques for EUS-Guided Fine-Needle Aspiration of Pancreatic Solid Masses at Facilities without On-Site Cytopathology: Results from Two Prospective Randomised Trials

**DOI:** 10.3390/jcm10204662

**Published:** 2021-10-12

**Authors:** Woo Hyun Paik, Joon Hyuk Choi, Yangsoon Park, Jung Bok Lee, Do Hyun Park

**Affiliations:** 1Department of Internal Medicine and Liver Research Institute, Seoul National University Hospital, Seoul National University College of Medicine, Seoul 03080, Korea; whpaik@snuh.org; 2Department of Internal Medicine, Inje University Haeundae Paik Hospital, Busan 48108, Korea; cladius@naver.com; 3Department of Pathology, University of Ulsan College of Medicine, Asan Medical Center, Seoul 05505, Korea; ysp@amc.seoul.kr; 4Department of Clinical Epidemiology and Biostatics, University of Ulsan College of Medicine, Asan Medical Center, Seoul 05505, Korea; jungboklee@amc.seoul.kr; 5Division of Gastroenterology, Department of Internal Medicine, University of Ulsan College of Medicine, Asan Medical Center, Seoul 05505, Korea; 6Digestive Diseases Research Center, University of Ulsan College of Medicine, Seoul 05505, Korea

**Keywords:** EUS, fine-needle aspiration, pancreatic neoplasm, suction

## Abstract

Background: EUS-guided fine-needle aspiration (EUS-FNA) has emerged as the primary modality for the cytologic diagnosis of pancreatic solid masses. The aim of this study is to determine whether technical factors including suction (S), non-suction (NS), capillary sampling with stylet slow-pull (CSSS), and the number of needle actuations (to-and-fro needle movements) may affect the accuracy of EUS-FNA for pancreatic solid masses at facilities without on-site cytopathology. Methods: The diagnostic yield of malignancy, blood contamination and cellularity at each sample acquired from EUS-FNA with or without S and different numbers of actuation (10, 15 and 20) were measured (study I). The optimal actuation number was determined and a head-to-head comparison trial between S and CSSS was performed (study II). Results: In study I, significant blood contamination was seen using S with 20 compared with 15 actuations (*p* = 0.002). Diagnostic yield of malignancy was not significantly different between 10, 15, and 20 actuations with S, whereas it was statistically higher for 15 actuations compared with 10 actuations with NS (*p* = 0.001). In study II, no difference was noted in diagnostic yield with 15 actuations between S and CSSS (88% vs. 90%, *p* = 0.74). Conclusions: Increasing actuation in NS resulted in a better diagnostic yield for EUS-FNA without significant blood contamination, whereas increasing actuation in S did not change the diagnostic yield of EUS-FNA while causing significant blood contamination. With 15 actuations, the diagnostic yield was comparable between S and CSSS.

## 1. Introduction

EUS-guided fine-needle aspiration (EUS-FNA) has emerged as the primary modality for the cytologic diagnosis of pancreatic solid masses due to its high accuracy and safety [1]. Several factors are important for the diagnostic accuracy of EUS-FNA, including the type and size of the needle [2,3,4], the number of needle passes [5], the presence of an on-site cytopathologist [6,7], the method of cytopathology preparation [8], the use of a stylet [9], the use and type of suction [10], fanning technique [11,12], and the use of contrast-enhanced EUS [13]. However, limited resources and financial constraints restrict the availability of on-site cytopathology in some centres. No standardized technique has yet been established for the use of EUS-FNA without on-site cytopathology; the accuracy of suction and needle actuation (to-and-fro needle movement) is debatable. [14] The European Society of Gastrointestinal Endoscopy (ESGE) recommended EUS-FNA with suction to improve sensitivity in solid lesions, even though it increases the amount of blood. A recent comparative study indicated that EUS-FNA using capillary sampling with minimal negative pressure, using a stylet slow pull technique, was associated with less blood contamination and could potentially increase the diagnostic yield of EUS-FNA for pancreatic solid masses when compared with EUS-FNA conducted with suction [15]. 

The use of needle actuation as a capillary action for sampling tissue generally involves 10–20 times performed during EUS-FNA with suction (S); however, the optimal number of actuations to perform during non-suction (NS) or capillary sampling with stylet slow-pull (CSSS) has not been established. Theoretically, a higher number of actuation would increase the risk of adverse events due to cellular damage [11]. Moreover, insufficient cellularity or increased blood contamination by low or excessive number of actuations could influence the total number of needle passes needed for a diagnosis in centres without an on-site cytopathologist. We determined whether the actuation with suction or without suction during EUS-FNA could affect the accuracy of pancreatic solid masses by conducting two prospective randomised trials in a centre without an on-site cytopathologist. The primary outcome of study I was the determination of the optimal number of actuations by comparing the diagnostic yield of EUS-FNA and the quality of the obtained specimen with and without suction. The objective of study II was to compare the diagnostic yield between S and CSSS methods with the determined optimal number of actuations determined in study I.

## 2. Patients and Methods

### 2.1. Patients

A total of 239 consecutive patients who underwent EUS-FNA conducted by a single experienced endoscopist (D.H.P.) for cytopathologic diagnosis of pancreatic solid mass were prospectively enrolled at a single tertiary hospital. This study was registered as a clinical trial (Clinical trial registration number: NCT01576497 in study I and NCT01923883 in study II). This endosonographer had previously reported an acceptable diagnostic accuracy (87% of positive cytology) of pancreatic solid masses in EUS-FNA without an on-site cytopathologist prior to commencement of this study [16]. The inclusion criteria were as follows: (1) age older than 18 years; (2) identifiable pancreatic solid lesions by spiral computed tomography (CT) scan; and (3) patients with informed consent. The exclusion criteria were as follows: (1) a complete cystic lesion without solid portion; (2) patients with bleeding tendency (platelet count <50,000/mm^3^ and/or prothrombin time international normalized ratio >1.5); and (3) refusal to participate in this study. Finally, 94 patients with suction and 99 patients without suction were enrolled and analysed (Figure 1). Patients were monitored by inpatient observation in terms of procedure-related adverse events during and after EUS-FNA. The complete blood count, liver function test, serum amylase and lipase levels were checked 24 h after the procedure. The Institutional Review Board of Asan Medical Centre approved this study (study I: IRB number 2012-0018, and study II: 2013-0574).

The final diagnosis of malignancy was defined by any of the following criteria: (1) malignant cytology at EUS-FNA; (2) malignant histopathology obtained by other means (e.g., surgery, ERCP, or US/CT-guided biopsy); or (3) clinical and/or imaging follow-up that was consistent with a diagnosis of pancreatic cancer, such as clinical progression, metastasis, or death. If signs of malignancy were absent for more than 1 year, then the lesion was considered to be benign. 

### 2.2. Randomisation and Masking

Computer-generated randomisation assignments were obtained by the statistician before study enrolment. These were placed in sequentially numbered, sealed, opaque envelopes and opened by a nurse unrelated to this study during the procedure. Masking of investigators was undertaken in study I. Given the different technical characteristics of S and CSSS, the investigators were not masked in study II. Independent observers (W.H.P. and J.H.C.) who had no knowledge of the purpose of this study prospectively evaluated cytologic outcomes and other variables of EUS-FNA. Investigators (D.H.P. and Y.P.) as an operator and pathologist were unaware of the data analysis until the database was closed for the final analysis.

### 2.3. EUS-FNA Technique

Informed prior consent for EUS-FNA was obtained from all the patients. EUS-FNA was performed using a linear array echoendoscope (GF-UCT240 or 260, Olympus Optical Tokyo, Japan) under conscious sedation with midazolam and meperidine. The lesion size was measured across its longest diameter. The puncture was then performed targeting a periphery of the lesion using a 22-gauge needle (ECHO-3-22, Cook Endoscopy, Winston-Salem, NC, USA) guided by real-time EUS imaging. After the lesion was punctured, the stylet was completely removed, and suction was applied with a 10 mL syringe while oscillating the needle inside the lesion. Two types of suction syringe were applied: a true suction syringe and a false one (Appendix A). The syringes were randomly assigned and loaded by a nurse so that the operator was unaware of whether suction was being applied. The stopcock was rotated, and the suction or sham suction was turned off by an assistant nurse before withdrawing the needle from the lesion. The order of suction or non-suction was determined by a pre-printed randomisation sequence kept in an envelope that was opened by the assistant nurse after enrolment. At least three needle passes with 10, 15 and 20 actuations were conducted per lesion, based on our previous study [16]. The actuation involved jabbing or staccato needle movements for a capillary and shearing effect. Needle passes were taken from the periphery and the largest diameter of the lesion as possible.

If the obtained specimen seemed insufficient by gross inspection, an additional needle pass with 25 actuations was performed. If the specimen still appeared insufficient for pathologic exam by visual assessment, a fifth needle pass was done at a different site or by using a fanning technique with 20 actuations. The cytologic results of these fourth and fifth needle passes were not analysed in this study because the rescue method resulted in a different sample sizes and missing data for each patient. The inside of the needle was rinsed vigorously with normal saline after every needle puncture to prevent cross-contamination of needles or bleeding after the initial procedure.

In study II with CSSS, multiple actuations were performed within the target lesion with simultaneous minimum negative pressure provided by slowly and continuously pulling the needle stylet. The aspirated material was expelled onto glass slides by reinsertion of the stylet and flushing with air. Cytological smears were made for all aspirated specimens by endosonographers trained in cytological slide preparation techniques. Blinded results from the operator were ensured by not performing the on-site cytopathologic evaluations at the time of the EUS-FNA procedures. The adequacy of each obtained specimen was evaluated by gross inspection by the operator, as follows: Poor—no cellular material or inconspicuous whitish core mixed with blood; Fair—presence of whitish core; Good—whitish core with the presence of a wormlike tissue architecture [17]. These slides were labelled according to the number of needle passes. Smeared slide glasses were fixed in absolute alcohol and stained later with Papanicolau stain. A cell block exam was not performed in this study.

### 2.4. Cytological Interpretation

A single experienced cytopathologist (Y.P.), who was blinded to the use of suction and the number of actuations during EUS-FNA, interpreted the cellularity, bloodiness and cytologic diagnosis of each obtained specimen. The slides for each pass were assessed using strict predefined criteria for the following: cellularity, adequacy of specimen, amount of blood, and diagnosis. Cellularity was determined by the percentage of area of slide that contained cells of the representative lesion. The presence of cells on more than 25% of total area of slide was considered to be sufficient cellularity. Bloodiness was graded into 3 levels: Minimal—a few blood cells without affecting cytopathology diagnosis; Moderate—partially obscured by blood cells but a cytopathology diagnosis still possible; Significant —too obscured by blood cells for an adequate interpretation [18]. The cytological samples were categorized as malignant, suspicious for malignancy, atypical, benign or inadequate according to the Papanicolaou Society of Cytopathology System for Reporting Pancreatobiliary Cytology [19]. The “suspicious for malignancy” represents the loss of cell polarity resulting in significant architectural disorder, significant nuclear enlargement, nuclear membrane irregularities, presence of a coarse chromatin pattern, distinct nucleoli and increased nuclear/cytoplasmic ratio. The “atypical” represents loss of architectural polarity with mild alterations and loss of the benign honeycomb pattern. “Malignant” and “suspicious for malignancy” were considered positive for malignancy.

### 2.5. Statistical Analysis

Sample size in study I was calculated based on previously published literature on the diagnostic accuracy of EUS-FNA with S and NS (88% and 67%) [20]. The expected difference of 20% required 91 patients in each group (S or NS), assuming a 5% significance level and statistical power of 90% tested against a 2-sided alternative.

In study II, we assumed that the diagnostic yield of malignancy in S and CSSS with optimal actuation was equivalent to 88%. The required sample size was calculated based on a margin of non-inferiority for a success rate of 0.1. A statistical power of 80% with the assumption of a one-sided type I error rate of 0.02 indicated that 46 patients were required in each group (S or CSSS). At the recommendation of a biostatistician, each patient underwent four instances of needle passes with the S or with CSSS method used alternatively and targeting same lesion. The order of S or CSSS was determined randomly.

Statistical analyses were performed using SPSS v.18.0 (IBM Corp., Armonk, NY, USA). Numerical data are presented as the median with the range or as the mean value with the standard deviation (SD). Intergroup comparisons for continuous variables were performed using Student’s *t*-test or the Mann–Whitney *U*-test. Categorical and binary variables were tested using the χ^2^ test or Fisher’s exact test. The correlations between the number of actuations and diagnostic outcomes were analysed using a generalized estimating equation. Since cytological results according to the number of actuations (10/15/20) were obtained in the same lesion with repeated measures, a generalized estimating equation was used. Estimated marginal means of positive results in significant blood contamination and diagnostic yield for the number of actuations were analysed with repeated contrast adjusted by a least significant difference method. The predictive factors affecting the diagnostic yield of EUS-FNA were determined by multivariate logistic regression performed per each needle pass. *p* values < 0.05 indicated statistical significance.

## 3. Results

### 3.1. Determination of Optimal Actuation Number (Study I)

Table 1 shows the baseline characteristics of enrolled patients. The total number of needle passes was 385 with suction (S) and 453 without suction (NS). Visual assessment by an endosonographer and without on-site cytologic evaluation revealed that the mean number of needle passes and the proportion of inadequate specimens were significantly more frequent in the NS group than in the S group (4.6 vs. 4.1, *p* = 0.01; 0.6 vs. 0.2, *p* = 0.001, respectively). Use of the generalized estimating equation model, in the per-lesion-based analysis, revealed statistical differences in the diagnostic yield of malignancy in each actuation of the NS group (10 [61%] vs. 15 [76%] actuations, *p* = 0.001; 10 [61%] vs. 20 [78%] actuations, *p* = 0.003). By contrast, no differences were found in the diagnostic yield of malignancy in each actuation of the S group (10 [90%] vs. 15 [86%], *p* = 0.26; 10 [90%] vs. 20 [92%] actuations, *p* = 0.71) (Table 2). The per-needle pass analysis revealed significant blood contamination for 20 actuations in the S group compared with 15 actuations (*p* = 0.002). No difference was observed in blood contamination for each actuation of the NS group (Table 3). The diagnostic yield of malignancy was statistically lower for 10 actuations in the NS group compared with 15 actuations (*p* = 0.001). No difference was indicated in the diagnostic yield of malignancy in the other actuations of the S (10 vs. 15, 15 vs. 20) and NS (15 vs. 20) groups (Table 3). The cellularity was similar with 10, 15 and 20 actuations in the S group (32/74 [43%] vs. 33/74 [45%] vs. 36/74 [49%], *p* = ns) and NS group (22/84 [26%] vs. 32/84 [38%] vs. 29/84 [35%], *p* = ns).

To determine the significant factors affecting the diagnostic accuracy of EUS-FNA, univariable analysis was done with the following factors: number of needle passes, size of the lesion, puncture site and suction. Among them, the size of the lesion and the puncture site were the factors affecting the diagnostic accuracy of EUS-FNA (Table 4). However, there was no significant factor determining the diagnostic accuracy of EUS-FNA in multivariable analysis.

### 3.2. A Head-to-Head Comparison between Suction and Capillary Suction (Study II)

Based on the above results, the assumed optimal number of actuations was determined as 15 times because the blood contamination was significantly lower with 15 actuations than with 20 actuations in the S group and the diagnostic yield was significantly higher with 15 actuations than with 10 actuations in the NS group. Therefore, we performed a head-to-head comparison and non-inferiority trial between the S and CSSS methods, setting the assumed optimal number of actuations as 15 times. At the recommendation of a biostatistician, each patient underwent needle pass four with or without suction alternatively targeting the same lesion (Figure 2). The order of suction or CSSS was determined randomly (S-CSSS-S-CSSS or CSSS-S-CSSS-S). The operator was blinded to the use of suction vs. sham suction under the blindness of the practitioner like the previous one.

A total of 184 needle passes were performed in 46 patients, such that 92 needle passes were done alternatively by using the S or CSSS method (Table 5). The cytologic adequacy rate of the specimen was significantly higher with suction than without suction (93% vs. 81%, *p* = 0.02, Table 6). The cellularity and blood contamination of specimen were also higher with suction than without suction (40% vs. 24%, *p* = 0.02; 39% vs. 19%, *p* < 0.01, respectively). No difference in the diagnostic yield of malignancy was seen in the suction and non-suction groups (87% vs. 80%, *p* = 0.15).

### 3.3. Adverse Events

No adverse events, needle malfunction or significant clogging related to the EUS-FNA procedure were reported.

## 4. Discussion

Although EUS-FNA is reported to be a safe and effective method for the diagnosis of pancreatic solid masses [21,22], an optimal technique for EUS-FNA has not been established. Several factors affect the accuracy of EUS-FNA; for example, the skill and experience of the endosonographer may influence the results of EUS-FNA [23,24,25]. In the present study, the participation of an experienced endosonographer for EUS-FNA may have minimised bias due to an operator factor [16,17]. Our study is the prospective randomised trial to examine the role of actuation with suction (S) vs. non-suction (NS) and capillary sampling with a stylet slow-pull (CSSS) method during EUS-FNA of pancreatic solid masses. During the first study, the endosonographer was blinded to the results of EUS-FNA because an on-site cytopathologist was not available and the use of suction was masked. Therefore, bias from the subjectivity of the operator could be avoided. The cytopathologist was also blinded to the use of suction and the number of actuations in each needle pass during the EUS-FNA procedure.

The first randomised trial (study I) for finding an optimal actuation indicated significant blood contamination in per needle pass analysis with 20 actuations in the S group compared with 15 actuations (*p* = 0.002). No difference was noted in blood contamination for any number of actuations in the NS group (Table 3). The diagnostic yield of malignancy was statistically lower for 10 actuations in the NS group compared with 15 actuations (*p* = 0.001). No difference was observed in the diagnostic yield of malignancy for the actuation of the S (10 vs. 15, 15 vs. 20) and NS (15 vs. 20) groups. These results suggested that the optimal number of actuations may be important in EUS-FNA without S but is not important in EUS-FNA with S in terms of the diagnostic yield of malignancy.

However, an excessive number of actuations is associated with blood contamination and cellular damage in EUS-FNA with S. The diagnostic accuracy of air-dried and Diff-Quick stain for immediate on-site interpretation, which can be affected by blood contamination when compared with ethanol-fixed and Papanicolaou stained samples, was not evaluated for the reduction of bias from subjectivity in the assessment of specimens by the cytopathologists. The finding that higher blood contamination was found with 20 actuations with S compared with 10 and 15 actuations indicated that EUS-FNA with more than 20 actuations under S may affect the accuracy of the FNA results at centres that have immediate on-site cytology interpretation available.

The mean number of needle passes and the proportion of inadequate specimens were significantly larger in the NS group than in the S group in study I (4.6 vs. 4.1, *p* = 0.01; 0.6 vs. 0.2, *p* = 0.001, respectively) because the number of inadequate specimens obtained with 10 actuations in the NS group was much higher, as determined by visual assessment. This may suggest that a suboptimal number of actuations in NS may be associated with a requirement for a greater number of needle passes in centres without on-site cytology evaluation.

The results of this study do not necessarily indicate that NS is related to a lower diagnostic yield. Study I employed a sham suction to mask the S or NS condition; however, the sham suction syringe is not the same as other trials without S where a syringe was not attached. The attachment of a closed syringe prevents the venting effect that occurs when no syringe is attached, which may facilitate sample acquisition by capillary action. This may have led to less adequate sampling and a lower diagnostic yield in the NS group. Therefore, we compared the S and CSSS methods in study II to preclude this effect of a closed syringe and also to verify the importance of optimal actuation under circumstances where samples are obtained by capillary sampling with minimal negative pressure. No difference was found in study II in the diagnostic yield of malignancy in either group (S, 88% vs. CSSS, 90%, *p* = 0.74). An optimal number of actuations during EUS-FNA gave a comparable diagnostic yield with either S or CSSS.

This study has several limitations. We used a 22-gauge EUS-FNA needle only for the evaluation of optimal actuation. Therefore, our results obtained for actuation with a 22-gauge needle cannot be generalized to actuation for a 25-gauge needle. However, a recent prospective randomised trial revealed that the diagnostic cell block yield of a 25-gauge needle was low without on-site cytopathology [26]. These findings indicate that a needle calibre larger than 25-gauge would be preferred in centres where an on-site cytopathologist is not available. We did not study differences in needle design to determine if this had an effect. Recently, EUS needles with side fenestration or specially designed tips, including SharkCore and Franseen, have been widely used for core biopsy [27,28,29,30]. Since EUS sampling is moving from FNA to core biopsy [31], prospective randomized clinical trials using novel core biopsy needles would be warranted. Cell block evaluation was not performed in this study, so a prospective study on the role of actuation and suction in a cell block without an on-site cytopathologist would be of interest.

Limitations related to study design should also be considered. Because only the use of suction or not, not the number of actuations, was randomized in study I, it seems that the quality of the specimens with 20 actuations, which was always performed as a third needle pass, could be affected by the first two passes. The accuracy rate of NS in study I was low because the NS which was applied in study I was a sham suction, and the syringe was prepared as the piston was pulled down in the open state of the valve and then the valve was closed later. Therefore, it was different from non-suction in real clinical practice because the side where the syringe is attached was completely blocked. Fifteen times of needle actuation, which was the optimal actuation number in the suction and non-suction group in study I, were applied in both suction and CSSS in study II for a head-to-head comparison. However, strictly speaking, only the optimal number of NS, not CSSS, was checked before conducting study II. A bench test showed CSSS had −2 inHg and 10 mL negative suction had −20.5 inHg (Appendix A, inHg: A unit of measure for pressure or vacuum, used to indicate how high a column of mercury can be pushed by pressure within a sealed tube). In study I, sham suction had no negative suction as seen in Appendix A. Therefore, the circumstance on pressure in the non-suction group (sham suction, 0 inHg) in study I may be similar to that in CSSS (−2 inHg). Thus, we choose 15 times of needle actuation for CSSS in study II. However, because the optimal number of actuations is not directly determined in the CSSS technique, further studies comparing suction with 15 actuations vs. CSSS with more than 15 actuations would be warranted.

The accuracy of fanning with multiple actuations may be superior to that of a standard EUS-FNA technique [12]. In our study, 10, 15, 20 and 25 actuations were performed at the same site (the periphery and longest diameter of mass) rather than fanning. The fanning technique was only used as a rescue EUS-FNA technique in the fifth needle pass, with 20 actuations. We designed the study protocol in this way to reduce the confounding bias which could result from the fanning technique.

This study was conducted without on-site cytopathology. The use of on-site cytopathology is recognised as a superior method for increasing diagnostic yield in EUS-FNA [32]. However, an on-site cytopathologist is not available in most institutions, due to added time and cost [33,34]. The absence of an on-site cytopathologist increases the importance of using an appropriate EUS-FNA technique to obtain an adequate cytologic specimen. A recent study reported no significant difference in the diagnostic yield of malignancy and the proportion of inadequate specimens in EUS-FNA with or without an on-site cytopathologist when EUS-FNA was performed by experienced endosonographers. Our findings indicate the importance of actuation under NS and CSSS conditions, which may affect the diagnostic yield of EUS-FNA in centres without on-site cytopathologists.

In conclusion, increasing the number of actuations under NS conditions appeared to yield a better diagnosis with EUS-FNA without significant blood contamination. By contrast, increasing the number of actuations under S conditions had no effect on the diagnostic yield of EUS-FNA but caused significant blood contamination. When 15 actuations are used, the diagnostic yield may be comparable for S and CSSS.

## Figures and Tables

**Figure 1 jcm-10-04662-f001:**
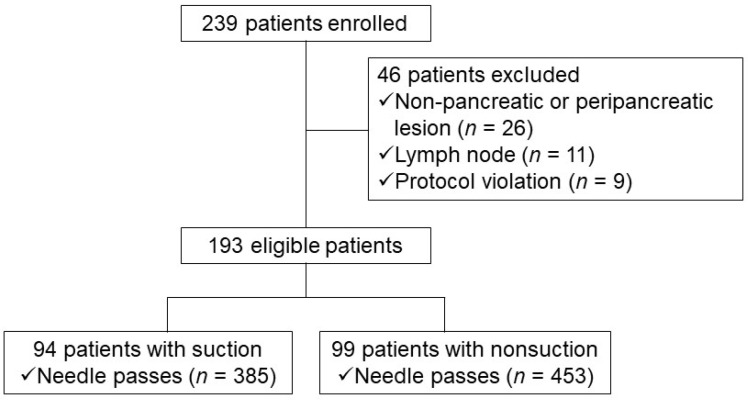
Study flow chart depicting patient selection (study I).

**Figure 2 jcm-10-04662-f002:**
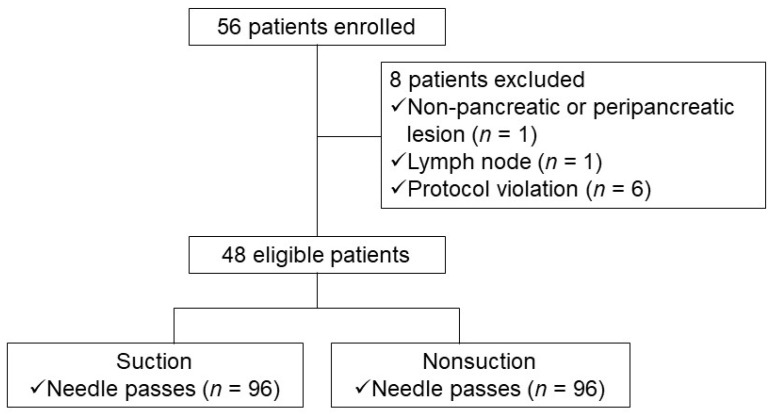
Study flow chart depicting patient selection (study II).

**Table 1 jcm-10-04662-t001:** Baseline characteristics of enrolled patients (study I).

	Suction (*n* = 94)	Non-Suction (*n* = 99)	*p*-Value
Age (mean ± SD, years)	60.0 ± 13.2	59.7 ± 12.3	0.86
Male to female ratio	66/28	58/41	0.10
Size of mass (Longest diameter, mm)	2.89 ± 1.21	3.00 ± 1.49	0.68
Puncture site			0.89
Stomach	45	50	
Duodenum	48	48	
Stomach and duodenum	1	1	
Number of needle passes (mean + SD)	4.1 ± 1.2	4.6 ± 1.4	0.01
Proportion of inadequate specimens	0.2	0.6	0.001
Final diagnosis			0.11
Pancreatic cancer	74 (78.7)	79 (79.1)	
Neuroendocrine tumour	9 (9.6)	6 (6.1)	
IPMN	4 (4.3)	1 (1.0)	
SPN	3 (3.2)	1 (1.0)	
Others	4 (4.3) *	12 (12.0) ^†^	

* Chronic pancreatitis (*n* = 3) and pancreatic abscess (*n* = 1); ^†^ Chronic pancreatitis (*n* = 11) and autoimmune pancreatitis (*n* = 1).

**Table 2 jcm-10-04662-t002:** Comparison of diagnostic yield according to the number of needle actuations (10/15/20) using suction and non-suction methods in malignant lesions (per lesion, study I).

Variables	Number of Actuations	Suction (74 Lesions)			Non-Suction (79 Lesions)		
		Positive Ratio (Number)	OR (95% CI)	*p*-Value	Positive Ratio (Number)	OR (95% CI)	*p*-Value
Diagnostic yield	10	90% (67/74)			61% (48/79)		
	15	86% (64/74)	0.66 (0.33–1.33)	0.26	76% (60/79)	2.15 (1.34–3.44)	0.001
	20	92% (68/74)	1.18 (0.49−2.84)	0.71	78% (62/79)	2.50 (1.37–4.54)	0.003

The reference value is 10 actuations, and the OR and *p*-values represent the comparison of 10 actuations with 15 or 20 actuations.

**Table 3 jcm-10-04662-t003:** Repeated contrast among the number of needle actuations (per needle pass, study I).

Variables	Comparison between Numbers of Actuations	*p*-Value	
		Suction	Non-Suction
Significant blood contamination	10 versus 15	0.10	0.86
	15 versus 20	0.002	0.71
	20 versus 25	>0.99	0.41
Diagnostic yield	10 versus 15	0.25	0.001
	15 versus 20	0.15	0.56
	20 versus 25	>0.99	>0.99

**Table 4 jcm-10-04662-t004:** Univariable and multivariable analysis about the factors affecting the diagnostic accuracy of EUS-guided fine-needle aspiration.

Variables	No. (%) of Successful Diagnosis	*p*	Multivariable Analysis	
			OR (95% CI)	*p*
Size of lesion (cm)		0.045	2.25 (0.84−6.06)	0.11
<3 (*n* = 108)	90 (83%)			
≥3 (*n* = 85)	79 (93%)			
Approach		0.04	2.28 (0.88−5.88)	0.09
Transgastric (*n* = 95)	88 (93%)			
Transduodenal (*n* = 98)	81 (83%)			
Suction		0.46		
Yes (*n* = 94)	84 (89%)			
No (*n* = 99)	85 (86%)			
Needle pass number		0.56		
1−3 (*n* = 70) ≥4 (*n* = 123)	60 (86%) 109 (89%)			

**Table 5 jcm-10-04662-t005:** Baseline characteristics of enrolled patients in a head-to-head comparison study between suction and non-suction (study II).

Age (mean ± SD, years)	61.3 ± 10.2
Male to female ratio (M:F)	21:27
Size of mass (long diameter, mean ± SD, mm)	2.66 ± 1.24
Tumour location, *N* (%)	
Uncinate process	2 (4.2)
Head	24 (50.0)
Neck	4 (8.3)
Body	7 (14.6)
Tail	11 (22.9)
Puncture site, *N* (%)	
Stomach	22 (45.8)
Duodenum	26 (54.2)
Stomach and duodenum	0
Final diagnosis, *N* (%)	
Pancreatic cancer	39 (81.3)
Neuroendocrine tumour	7 (14.6)
Solid pseudopapillary neoplasm	1 (2.1)
Chronic pancreatitis	1 (2.1)

**Table 6 jcm-10-04662-t006:** Cytological results of samples that were obtained with suction or capillary sampling with stylet slow-pull (CSSS) in the same lesion using the optimal number of actuations (per needle passes, study II).

Parameter	Suction (*n* = 96)	CSSS (*n* = 96)	*p*-Value
Adequacy of specimen			0.02
Inadequate	7 (7.3)	18 (18.8)	
Adequate	89 (92.7)	78 (81.3)	
Amount of blood			<0.01
Minimal	23 (24.0)	42 (43.8)	
Moderate	36 (37.5)	36 (37.5)	
Significant	37 (38.5)	18 (18.8)	
Cellularity			
% Of area of slide that contains cells of the representative lesion			0.02
No representative cells present	7 (7.3)	15 (15.6)	
Representative cells present in <25%	51 (53.1)	58 (60.4)	
Representative cells present in 25–50%	25 (26.0)	19 (19.8)	
Representative cells present in >50%	13 (13.5)	4 (4.2)	
Diagnosis			0.24
Benign or others	13 (13.5)	10 (10.4)	
Atypical	7 (7.3)	7 (7.3)	
Suspicious	11 (11.5)	12 (12.5)	
Malignant	57 (59.4)	50 (52.1)	
Inadequate for reporting	8 (8.3)	17 (17.7)	

## Data Availability

The data presented in this study are available on request from the corresponding author.

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
