# Peer review of "Optimal Techniques for EUS-Guided Fine-Needle Aspiration of Pancreatic Solid Masses at Facilities without On-Site Cytopathology: Results from Two Prospective Randomised Trials"

_jcm, 2021, doi:10.3390/jcm10204662_

Round 1
Reviewer 1 Report
Comment to the Authors
This is a study of EUS-FNA with suction, non-suction and a stylet slow-pull method. This is a very interesting subject for an endosonographer, and I have a number of queries for the authors.
- Isn't the description of Page5, Line207-208 described in the table?
- Please describe the difference between the definitions of NS and CSSS.
- I think that the accuracy rate of NS in Study I is low compared to the previous reports. What do you think is the cause?
- There is no explanation about the P-value in Table 2. Please describe which 2 samples were tested in the table.
Author Response
Reviewer #1
Comment to the Authors
This is a study of EUS-FNA with suction, non-suction and a stylet slow-pull method. This is a very interesting subject for an endosonographer, and I have a number of queries for the authors.
- Isn't the description of Page5, Line207-208 described in the table?
- Many thanks for your kind comments. We added the results regarding the mean number of needle passes and the proportion of inadequate specimen between NS and S groups in Table 1 as follows:
Table 1. Baseline characteristics of enrolled patients (study I)
|
|
Suction (n = 94) |
Nonsuction (n = 99) |
P-value |
|
Age (mean ± SD, years) |
60.0 ± 13.2 |
59.7 ± 12.3 |
.86 |
|
Male to female ratio |
66/28 |
58/41 |
.10 |
|
Size of mass (longest diameter, mm) |
2.89±1.21 |
3.00±1.49 |
.68 |
|
Puncture site |
|
|
.89 |
|
Stomach |
45 |
50 |
|
|
Duodenum |
48 |
48 |
|
|
Stomach and duodenum |
1 |
1 |
|
|
Number of needle passes (mean + SD) |
4.1 ± 1.2 |
4.6 ± 1.4 |
.01 |
|
Proportion of inadequate specimens |
0.2 |
0.6 |
.001 |
|
Final diagnosis |
|
|
.11 |
|
Pancreatic cancer |
74(78.7) |
79(79.1) |
|
|
Neuroendocrine tumor |
9(9.6) |
6(6.1) |
|
|
IPMN |
4(4.3) |
1(1.0) |
|
|
SPN |
3(3.2) |
1(1.0) |
|
|
Others |
4(4.3)* |
12(12.0)† |
|
- Pease describe the difference between the definitions of NS and CSSS.
- In study 1 with NS, sham suction syringe was prepared as the the piston was pulled down in the open state of valve and then the valve was closed later. In study 2 with CSSS, minimum negative pressure was provided by slowy and continuously pulling the needle stylet. In order to help the reader understand, the explanation of NS was additionally written as a supplementary.
- I think that the accuracy rate of NS in Study I is low compared to the previous reports. What do you think is the cause?
- Many thanks for your important comments. The NS which was applied in study I was a sham suction, and the syringe was prepared as the piston was pulled down in the open state of valve and then the valve was closed later. Therefore, it was different from non-suction in clinical practice because the side where the syringe is attached was completely blocked. This might contribute to the low accuracy rate. We mentioned this in DISCUSSION section as follows:
“The accuracy rate of NS in study I was low because the NS which was applied in study I was a sham suction, and the syringe was prepared as the piston was pulled down in the open state of valve and then the valve was closed later. Therefore, it was different from non-suction in real clinical practice because the side where the syringe is attached was completely blocked.”
We also explained about sham suction as supplementary figure.
“Supplementary Figure 1. Two types of 10 mL suction syringes were applied randomly and the operator was blind to whether suction was used or not during EUS-guided fine needle aspiration. (A) True suction syringe was prepared as the piston was pulled down in the closed state of valve. (B) Sham suction (nonsuction) syringe was prepared as the the piston was pulled down in the open state of valve and then the valve was closed later.”
Suppl Fig. 1A
Suppl Fig. 1B
- There is no explanation about the P-value in Table 2. Please describe which 2 samples were tested in the table.
- The reference value in Table 2 is 10 actuations, and the OR and p-values represent the comparison of 10 actuations with 15/20 actuations. We annotated this in Table 2.

Reviewer 2 Report
This prospective study is composed of two randomized trials, aimed at assessing the optimal number of to-and-fro needle movements during EUS-FNA and, subsequently, the best technique between slow-pull and suction. This is an interesting study, well-conducted, and well-written. I have some comments and suggestions for the Authors:
- In study number I, you aimed at evaluating the optimal number of actuations. However, you randomized the use of suction and not the number of to-and-fro movements. Therefore, it is possible that the quality of the specimens of passes with 20 movements (performed always as a third pass) could be affected by the first two passes. This point represents a limitation to the study and should be mentioned in the discussion.
METHODS:
- you excluded “cystic lesions”. What about mixed/complex solid/cystic lesions? Please clarify the exclusion criteria.
- because you are evaluating the accuracy of EUS-FNA, EUS-FNA should be excluded from the criteria to assess the final diagnosis.
- Please specify the shorter time of follow-up to assess malignancy or benign lesion in non-resected patients (6 months? 12 months?)
- Please refer to the Papaniculaou classification when talking about cytological interpretation (Pitman MB, Layfield L. The Papanicolaou Society of Cytopathology system for reporting pancreaticobiliary cytology. Switzerland: Springer International Publishing, 2015:6).
- It is of paramount importance to clearly state how “suspicious for malignancy” and “atypical” cytological results were considered.
Minor:
- INTRODUCTION: In the second sentence you correctly stated factors associated with diagnostic accuracy. You should add some citation for each factor: needle type (PMID: 32433914, PMID: 33390343) and needle size (PMID: 29026598), number of needle passes (PMID: 28025154), presence of an on-site cytopathologist (PMID: 34116031, PMID: 26346868), the method of cytopathology preparation (PMID: 30484917), the use of stylet (PMID: 29535060), the use and type of suction (PMID: 32355882). Moreover, the use of contrast-enhanced EUS should be also mentioned (PMID: 33481633)
- Page 2 line 98: please spell CBC
- A deeper comment about the fact that EUS sampling is moving from FNA to FNB is needed in the discussion section. Doing so, cite PMID: 29941722
Author Response
Reviewer #2
This prospective study is composed of two randomized trials, aimed at assessing the optimal number of to-and-fro needle movements during EUS-FNA and, subsequently, the best technique between slow-pull and suction. This is an interesting study, well-conducted, and well-written. I have some comments and suggestions for the Authors:
- In study number I, you aimed at evaluating the optimal number of actuations. However, you randomized the use of suction and not the number of to-and-fro movements. Therefore, it is possible that the quality of the specimens of passes with 20 movements (performed always as a third pass) could be affected by the first two passes. This point represents a limitation to the study and should be mentioned in the discussion.
- Many thanks for your important comments. We totally agree with your opinion, and we mentioned this as a limitation of this study in the Discussion section as follows:
“Limitations related to study design should also be considered. Because only use of suction or not, not the number of actuations, was randomized in study I, it seems that the quality of the specimens with 20 actuations, which was always performed as a third needle pass, could be affected by the first two passes.”
METHODS:
- You excluded “cystic lesions”. What about mixed/complex solid/cystic lesions? Please clarify the exclusion criteria.
- Complete cystic lesions without solid portion were excluded in this study, and mixed solid/cystic lesions were included. According to the reviewer’s suggestion, we revised as follows:
“The exclusion criteria were as follows: 1) a complete cystic lesion without solid portion”
- Because you are evaluating the accuracy of EUS-FNA, EUS-FNA should be excluded from the criteria to assess the final diagnosis.
- Because the risk of false positive for malignancy is extremely low in EUS-FNA, a positive EUS-FNA cytological interpretation for malignancy in patients with unsuitable for surgical resection can provide proof as to the presence of a malignancy. In previous study (Clin Gastroenterol Hepatol 2017;15:1071-1078), the diagnosis of malignant cells at EUS-FNA was also included in the final diagnosis.
- Please specify the shorter time of follow-up to assess malignancy or benign lesion in non-resected patients (6 months? 12 months?)
- Many thanks for your kind comment. If signs of malignancy were absent for more than 1 year, then the lesion was considered to be benign. We mentioned this in Methods section.
- Please refer to the Papaniculaou classification when talking about cytological interpretation (Pitman MB, Layfield L. The Papanicolaou Society of Cytopathology system for reporting pancreaticobiliary cytology. Switzerland: Springer International Publishing, 2015:6).
- Many thanks for your kind comment. The cytological samples were interpreted according to the Papanicolaou Society of Cytopathology System for Reporting Pancreatobiliary Cytology. We mentioned this in METHODS section.
- It is of paramount importance to clearly state how “suspicious for malignancy” and “atypical” cytological results were considered.
- We addressed “suspicious for malignancy” and “atypical” cytological results in METHODS section as follows:
“The “suspicious for malignancy” represents the loss of cell polarity resulting in significant architectural disorder, significant nuclear enlargement, nuclear membrane irregularities, presence of a coarse chromatin pattern, distinct nucleoli, and increased nuclear/cytoplasmic ratio. The “atypical” represents loss of architectural polarity with mild alterations and loss of the benign honeycomb pattern.”
Minor:
- INTRODUCTION: In the second sentence you correctly stated factors associated with diagnostic accuracy. You should add some citation for each factor: needle type (PMID: 32433914, PMID: 33390343) and needle size (PMID: 29026598), number of needle passes (PMID: 28025154), presence of an on-site cytopathologist (PMID: 34116031, PMID: 26346868), the method of cytopathology preparation (PMID: 30484917), the use of stylet (PMID: 29535060), the use and type of suction (PMID: 32355882). Moreover, the use of contrast-enhanced EUS should be also mentioned (PMID: 33481633)
- Many thanks for your kind comments. We cited above references according to the comments as follows:
“Several factors are important for the diagnostic accuracy of EUS-FNA, including the type and size of needle [2-4], the number of needle passes [5], the presence of an on-site cytopathologist [6,7], the method of cytopathology preparation [8], the use of stylet [9], the use and type of suction [10], fanning technique [11,12], and the use of contrast-enhanced EUS [13].”
- Page 2 line 98: please spell CBC
- We corrected CBC to “complete blood count”.
- A deeper comment about the fact that EUS sampling is moving from FNA to FNB is needed in the discussion section. Doing so, cite PMID: 29941722
- Thanks for your suggestion. We cited and mentioned the trend of moving from FNA to FNB in DISCUSSION section as follows:
“Since EUS sampling is moving from FNA to core biopsy [31], prospective randomized clinical trials using novel core biopsy needles would be warranted”

Reviewer 3 Report
This is an article on optimal techniques for EUS-guided fine needle aspiration of pancreatic solid masses. This article contains some serious problems.
As written in the introduction section, the primary outcome of study 1 was to determine the optimal number of actuations by comparing the diagnostic yields of EUS-FNA and the quality of the obtained specimen with and without suction (lines 76-78). In this context, readers wonder which was superior: suction or without suction? However, diagnostic yields were not compared between suction and without suction, only compared among actuation numbers in each group of suction and without suction.
It is written in the results section that the predictive factors affecting the diagnostic yield of EUS-FNA for pancreatic solid masses were determined size of 3cm or more, target lesion located in pancreatic body or tail, with suction, and actuations of 15 or more were the significant factors affecting diagnostic yield of EUS-FNA in both univariate and multivariate analysis (lines 225-229). However, according to Table 4, they were only size and approach in univariate analysis, and neither of them were significant (P<0.05) in multivariate analysis. There are no data of location and actuation in Table 4. In the results section, different ORs and 95% CIs are written (lines 225-229) from those in Table 4.
In study 2, diagnostic yields of 15 actuations were compared between suction and CSSS. Actuation number of 15 was adopted according to the result of study 1 that compared diagnostic yield and blood contamination among 10, 15, and 20 actuations. However, there are no reasons why 15 actuations was optimal in CSSS. Optimal actuation number of CSSS should have been determined beforehand.
Author Response
Reviewer #3
This is an article on optimal techniques for EUS-guided fine needle aspiration of pancreatic solid masses. This article contains some serious problems.
- As written in the introduction section, the primary outcome of study 1 was to determine the optimal number of actuations by comparing the diagnostic yields of EUS-FNA and the quality of the obtained specimen with and without suction (lines 76-78). In this context, readers wonder which was superior: suction or without suction? However, diagnostic yields were not compared between suction and without suction, only compared among actuation numbers in each group of suction and without suction.
- In study I, we tried to find out the optimal number of needle actuations in each method (suction and nonsuction). In suction group, 15 or 20 actuations were not superior to 10 actuations regarding diagnostic yield, however, the blood contamination was significantly lower with 15 actuations than with 20 actuations. In nonsuction group, 15 or 20 actuations were superior to 10 actuations regarding diagnostic yield. Therefore, we determined the optimal number of actuations as 15 times. Because suction and nonsuction could not be compared head-to-head in study I, study II was additionally conducted.
- It is written in the results section that the predictive factors affecting the diagnostic yield of EUS-FNA for pancreatic solid masses were determined size of 3cm or more, target lesion located in pancreatic body or tail, with suction, and actuations of 15 or more were the significant factors affecting diagnostic yield of EUS-FNA in both univariate and multivariate analysis (lines 225-229). However, according to Table 4, they were only size and approach in univariate analysis, and neither of them were significant (P<0.05) in multivariate analysis. There are no data of location and actuation in Table 4. In the results section, different ORs and 95% CIs are written (lines 225-229) from those in Table 4.
- Many thanks for your important comments. The errors in Results mentioned by reviewer have been corrected as follows:
“To determine the significant factors affecting the diagnostic accuracy of EUS-FNA, univariable analysis was done with the following factors: number of needle pass, size of lesion, puncture site and suction. Among them, the size of lesion and puncture site was the factors affecting diagnostic accuracy of EUS-FNA (Table 4). However, there was no significant factor determining the diagnostic accuracy of EUS-FNA in multivariable analysis.”
- In study 2, diagnostic yields of 15 actuations were compared between suction and CSSS. Actuation number of 15 was adopted according to the result of study 1 that compared diagnostic yield and blood contamination among 10, 15, and 20 actuations. However, there are no reasons why 15 actuations was optimal in CSSS. Optimal actuation number of CSSS should have been determined beforehand.
- Many thanks for your comments. In Study II, 15 times of needle actuations were applied in CSSS for head-to-head comparison. However, strictly speaking as the reviewer advised, only the optimal number of NS, not CSSS, was checked before conducting study II. This could be a one of limitation related with study design, but referring to the results of study II, there seems to be no problem in CSSS setting the actuation number to 15. In addition, a bench test showed CSSS had -2 in.Hg and 10ml negative suction had -20.5 in.Hg in supplementary data (in.Hg: A unit of measure for pressure or vacuum, used to indicate how high a column of mercury can be pushed by pressure within a sealed tube). In study I, sham syringe suction had no negative suction as supple. Figure1. Therefore, the circumstance on pressure in non-suction group (sham suction, 0 in.Hg) in study I may be similar to that in CSSS (-2 in.Hg). Thus, we choose 15 times of needle actuation for CSSS in study II. We added this as supplementary data and acknowledged as follows. “Permission for use granted by Cook Endoscopy, Winston-Salem, NC, USA”. We mentioned this as a limitation to the DISCUSSION section as follows.
“Fifteen times of needle actuation, which was the optimal actuation number in the suction and nonsuction group in study I, were applied in both suction and CSSS in study II for head-to-head comparison. However, strictly speaking, only the optimal number of NS, not CSSS, was checked before conducting study II. A bench test showed CSSS had -2 in.Hg and 10ml negative suction had -20.5 in.Hg (Supplementary Figure 2, in.Hg: A unit of measure for pressure or vacuum, used to indicate how high a column of mercury can be pushed by pressure within a sealed tube). In study I, sham suction had no negative suction as Supplementary Figure 1. Therefore, the circumstance on pressure in non-suction group (sham suction, 0 in.Hg) in study I may be similar to that in CSSS (-2 in.Hg). Thus, we choose 15 times of needle actuation for CSSS in study II. Acknowledgement was added as follows: Permission for use of the bench test granted by Cook Endoscopy, Winston-Salem, NC, USA

Round 2
Reviewer 1 Report
The article had revied well.
Author Response
We appreciate your comments.
Reviewer 2 Report
Thank you for amending the manuscript.
I have one comment about your answer to my comment N6. I meant that you should clarify if "suspicious for malignancy" or "atypical" interpretations were considered "positive" or "negative" for malignancy. You can see this study (PMID 30654396) where these two different criteria were applied. Please, specify how you counted these cases in your study.
Author Response
Many thanks for your kind comments. We considered “malignant” and “suspicious for malignancy” were considered positive for malignancy, whereas “atypical” was not. We mentioned this in Methods section as follows:
“Malignant” and “suspicious for malignancy” were considered positive for malignancy.

Reviewer 3 Report
Unfortunately, this study is not well designed. It comprised two studies: study 1 which determined the optimal actuation numbers of suction and non-suction methods and study 2 which compared the diagnostic yield of suction and CSSS methods by the actuation number. However, in order to compare between suction and CSSS, was it necessary to determine the optimal actuation number of non-suction methods? In order to compare between suction and CSSS, authors should have determined the optimal actuation numbers of suction and CSSS method in study 1, not those of suction and non-suction methods.
Interpretation of the results of comparison between suction and non-suction methods is inconsistent. They said that accuracy rate of non-suction was low in the discussion section (page 11, paragraph 3, lines 3-7); however, they said that suction did not affect the diagnostic accuracy by univariate analysis in the results section (page 9, paragraph 2, lines 1-5). Finally, do authors recommend non-suction? This is not stated in the conclusions in abstract, nor mentioned in the last paragraph of the discussion section.
Authors wrote by mistake that size of 3cm or more, target lesion located in pancreatic body or tail, with suction, and actuations of 15 or more were the significant factors affecting diagnostic yield of EUS-FNA in both univariate multivariate analysis in the first draft. These results were rewritten according to my indication; however, they still say that size and location of the target lesion and the use of suction were other significant predictive factors determined to affect the diagnostic yield of EUS-FNA in the discussion section (page 10, paragraph 5, lines 1-2).
Author Response
Unfortunately, this study is not well designed. It comprised two studies: study 1 which determined the optimal actuation numbers of suction and non-suction methods and study 2 which compared the diagnostic yield of suction and CSSS methods by the actuation number. However, in order to compare between suction and CSSS, was it necessary to determine the optimal actuation number of non-suction methods? In order to compare between suction and CSSS, authors should have determined the optimal actuation numbers of suction and CSSS method in study 1, not those of suction and non-suction methods.
Thanks for your comments. The reason why nonsuction, instead of CSSS, was compared in study I was that the operator could perform EUS-FNA without knowing whether suction was applied or not. Since CSSS could not be performed blindly by the operator, the bias according to the operator's preference for suction or CSSS technique can be reflected. But, we totally agree with reviewer’s concerns, and further study comparing between suction with 15 actuations vs. CSSS with more than 15 actuations is required to determine the optimal actuation number in CSSS. The following sentence was added in Discussion.
“However, because the optimal number of actuation is not directly determined in CSSS technique, further studies comparing between suction with 15 actuations vs. CSSS with more than 15 actuations would be warranted.”
In addition, because the optimal number of actuation in CSSS was not fully evaluated, we removed the following sentence in abstract and conclusion, “Therefore endosonographers at centres without on-site cytopathology can choose either S with 10-15 actuation or CSSS with 15 actuation.” and “Therefore, an endosonographer in a centre without on-site cytopathology can choose either S and 10-15 actuations or CSSS and 15 actuations for evaluation of pancreatic solid masses using 22-gauge EUS-FNA.”
Interpretation of the results of comparison between suction and non-suction methods is inconsistent. They said that accuracy rate of non-suction was low in the discussion section (page 11, paragraph 3, lines 3-7); however, they said that suction did not affect the diagnostic accuracy by univariate analysis in the results section (page 9, paragraph 2, lines 1-5). Finally, do authors recommend non-suction? This is not stated in the conclusions in abstract, nor mentioned in the last paragraph of the discussion section.
Many thanks for your important comments. In study I, the number of needle passes was significantly higher in the nonsuction group than in the suction group. In fourth and fifth needle passes, 25 times of needle actuations or fanning technique was applied. Therefore, although suction did not affect the diagnostic accuracy in univariable analysis, it is difficult to conclude that suction did not affect the diagnostic accuracy because it was analyzed according to per lesion, not each needle pass. The aim of study I was not to evaluate whether suction or nonsuction is superior, but to determine the optimal number of actuation in each situation. As we mentioned in Discussion, the key point of study I is that 15 or more actuations may be required in EUS-FNA without suction, but more than 15 actuations resulted in more blood contamination without increasing diagnostic yield in EUS-FNA with suction.
Authors wrote by mistake that size of 3cm or more, target lesion located in pancreatic body or tail, with suction, and actuations of 15 or more were the significant factors affecting diagnostic yield of EUS-FNA in both univariate multivariate analysis in the first draft. These results were rewritten according to my indication; however, they still say that size and location of the target lesion and the use of suction were other significant predictive factors determined to affect the diagnostic yield of EUS-FNA in the discussion section (page 10, paragraph 5, lines 1-2).
Thanks for your comments. We removed this paragraph in Discussion.
